# First Examples of *s*-Metal Complexes with Subporphyrazine and Its Phenylene-Annulated Derivatives: DFT Calculations

**DOI:** 10.3390/ijms25136897

**Published:** 2024-06-24

**Authors:** Denis V. Chachkov, Oleg V. Mikhailov, Georgiy V. Girichev

**Affiliations:** 1Kazan Department of Joint Supercomputer Center of Russian Academy of Sciences–Branch of Federal Scientific Center “Scientific Research Institute for System Analysis of the RAS”, Lobachevskii Street 2/31, 420111 Kazan, Russia; de2005c@gmail.com; 2Department of Analytical Chemistry, Certification and Quality Management, Kazan National Research Technological University, K. Marx Street 68, 420015 Kazan, Russia; olegmkhlv@gmail.com; 3Department of Physics, Ivanovo State University of Chemistry and Technology, Sheremetevskiy Avenue 7, 153000 Ivanovo, Russia

**Keywords:** beryllium, subporphyrazine, [benzo]subporphyrazine, subphthalocyanine, macrocyclic complex, DFT method

## Abstract

Using quantum chemical calculation data obtained by the DFT method with the B3PW91/TZVP and M062X/def2TZVP theory levels, the possibility of the existence of four Be(II) coordination compounds, each of which contains in the inner coordination sphere and the double deprotonated forms of subporphyrazine (H_2_**SP**), mono[benzo]subporphyrazine (H_2_**MBSP**), di[benzo]subporphyrazine (H_2_**DBSP**), and tri[benzo]subporphyrazine (subphthalocyanine) (H_2_**TBSP**) with a ratio Be(II) ion/ligand = 1:1, were examined Selected geometric parameters of the molecular structures of these (666)macrotricyclic complexes with closed contours are given; it was noted that BeN3 chelate nodes have a trigonal–pyramidal structure and exhibit a very significant (almost 30°) deviation from coplanarity; however, all three 6-membered metal-chelate and three 5-membered non-chelate rings in each of these compounds are practically planar and deviate from coplanarity by no more than 2.5°. The bond angles between two nitrogen atoms and a Be atom are equal to 60° (in the [Be**SP**] and [Be**TBSP**]) or less by no more than 0.5° (in the [Be**MBSP**] and [Be**DBSP**]). The presence of annulated benzo groups has little effect on the parameters of the molecular structures of these complexes. Good agreement between the structural data obtained using the above two versions of the DFT method was noticed. NBO analysis data for these complexes are presented; it was noted that, according to both DFT methods used, the ground state of the each of complexes under study is a spin singlet. Standard thermodynamic parameters of formation (standard enthalpy Δ*_f_H*^0^, entropy *S*^0^, and Gibbs free energy Δ*_f_G*^0^) for the above-mentioned macrocyclic compounds were calculated.

## 1. Introduction

It is well known that modern chemical science, when searching for new substances with given properties, increasingly uses quantum chemical calculations, and only then comes the stage of the synthesis of a new substance. This work is devoted to the search for new macroheterocyclic compounds based on pyrrole with a reduced coordination cavity and a complex metal-free from extraligation.

Various derivatives of porphyrazine (see Figure 1a), as well as their numerous coordination compounds, in which they act as (NNNN)-donor atomic ligands due to their unique physicochemical properties and wide possibilities of their application in various branches of science and technology, have been considered at this point in time in a very large number of publications, the number of which is at least tens of thousands (see, in particular, review articles [1,2,3,4] and monographs [5,6]). The coordination compounds formed by them belong to the number of macrocyclic metal complexes with a closed loop and contain four articulated six-membered metal chelate cycles; at present, such compounds are known for almost all those *s-*, *p-*, *d*-, and *f*-elements that modern chemists generally deal with (the only exception is trans-actinides with atomic number Z > 110, obtained so far in negligible small quantities). Of no less interest is the structural analog of porphyrazine, subporphyrazine (Figure 1b), various derivatives of which are also devoted to a significant number of works, in particular [7,8,9,10,11,12,13,14,15,16,17,18,19,20,21,22,23,24,25]. Subporphyrazine and its substitutes, as well as porphyrazine, contain donor nitrogen atoms (although their number is smaller than porphyrazine) due to which, in principle, like porphyrazine, they are able to form macrocyclic coordination compounds with a closed loop, which should contain three articulated six-membered metal chelate cycles. However, at present, for some derivatives of subporphyrazine (in particular, subphthalocyanine and its analogs), coordination compounds with only one single complexing ion, namely B(III), are known [10,11,12,13,16,17,18,22,24]. A number of researchers attribute this circumstance to the small size of the “chelate cell” of subporphyrazine and its derivatives, as a result of which, only B(III) with a radius of only 23 pm can enter it. Among the doubly charged M(II) metal ions, the Be(II) ion (35 pm) has the smallest size, and one can hope for the existence of a complex of this ion with both subporphyrazine and its derivatives. Nevertheless, reliable experimental data confirming this prediction, as far as the authors of these lines are aware, have not been obtained up to now. In connection with this noted fact, it seems important to confirm or refute the existence of the Be(II) complex with subphthalocyanine using modern methods of quantum chemical calculation, namely, the methods of density functional theory (DFT). In this regard, the first task of this article will be the calculation of the molecular and electronic structures of the above complex. Along with this, it seems reasonable to carry out a similar calculation in parallel for Be(II) metal complexes with subporphyrazine (further H_2_**SP**) and derivatives close to it, namely, with mono[benzo]subporphyrazine H_2_**MBSP**, di[benzo]subporphyrazine H_2_**DBSP**, and tri[benzo]subporphyrazine H_2_**TBSP** (subphthalocyanine) containing annelated to subporphyrazine of the [benzo] group, the structural formulas of which are presented in Figure 2. 

Within the framework of this, it also seems useful to trace the role and influence of the number of annelated benzo groups on the parameters of molecular and electronic structures (in particular, on the size of the “chelate cells” of the above ligands), as well as on standard thermodynamic parameters (standard enthalpy (Δ*_f_H*^0^, standard entropy *S*^0^, and standard Gibbs free energy Δ*_f_G*^0^) of these complexes. All these issues will be discussed in the given article.

## 2. Results and Discussion

According to the data of each of the two DFT quantum chemical methods used by us, for the 2*s-*element under consideration, the formation of all four types of complexes mentioned above takes place. The lengths of chemical bonds between atoms and bond angles for [Be**SP**], [Be**MBSP**], [Be**DBSP**], and [Be**TBSP**] coordination compounds under consideration calculated by each of the given DFT methods are presented in Table 1. The images of the molecular structures of the given compounds obtained by the DFT B3PW91/TZVP method are shown in Figure 1. The images of the molecular structures obtained by the M06/def2TZVP method are similar to ones obtained by B3PW91/TZVP; they are shown in Appendix A. As can be seen in Table 1, both variants of the DFT method, DFT B3PW91/TZVP and DFT M062X/def2TZVP, not only predict stable molecular structure for each of these four complexes but also show that quantitatively, the parameters of their molecular structures differ only slightly from each other.

As can be seen in the data presented in Table 1, the Be–N bond lengths are practically independent of the nature of the subporphyrazine derivative and are in the range of 153.2–154.1 pm. Based on the structural formulas of the complexes we are considering, one could theoretically expect that in [Be**SP**] and [Be**TBSP**], the lengths of all three of these bonds would be the same, while in [Be**MBSP**] and [Be**DBSP**], only two bonds would be the same in length; the third will differ from the other two, and the calculation results are in full agreement with such a prediction (Table 1). A relatively small dependence on the nature of the macrocyclic ligand and the calculation method also takes place for the carbon–nitrogen and carbon–carbon bond lengths. More interesting is the situation with the bond angles between the atoms in the BeN3 chelate node and the six-membered metal chelate rings. Let us immediately note that, according to the data of both DFT B3PW91/TZVP and DFT M062X/def2TZVP, BeN3 chelate nodes in all these complexes have a trigonal–pyramidal structure with the sum of three angles.

(NBeN) is in the range of 332.4–333.0 (within the DFT B3PW91/TZVP method) and the range 330.6–330.9 (within the DFT M062X/def2TZVP method), i.e., it has a very significant (almost 30°) deviation from coplanarity (and, namely, from the sum of internal angles in a flat quadrangle equal to 360.0°). Since, as already mentioned above, the lengths of the beryllium–nitrogen bonds in the [Be**SP**] and [Be**TBSP**] complexes are the same, while in the [Be**MBSP**] and [Be**DBSP**] complexes they are different, it should be expected that the same will take place for bond angles, including the BeN3 chelate nodes, and this indeed takes place in the framework of the calculation by both the DFT B3PW91/TZVP method and the DFT M062X/def2TZVP method (Table 1). The values of these angles quite noticeably (by 9.0° and more) differ from 120.0°. The degree of deviation from coplanarity decreases somewhat in the series [Be**SP**]–[Be**MBSP**]–[Be**DBSP**]–[Be**TBSP**], although the size of the macrocycle chelate cell (which corresponds to the maximum diameter of a circle that can be inscribed in a triangle formed by nitrogen atoms bonded to the Be atom) practically does not change in this series (for example, according to the DFT B3PW91/TZVP method, it is 146.1 pm in [Be**SP**], 146.3 pm in [Be**MBSP**], 146.1 pm in [Be**DBSP**], and 146.0 pm in [Be**TBSP**]). (Moreover, in [Be**TBSP**], where the distances between the indicated nitrogen atoms are 253.0 pm, it is even slightly smaller than in [Be**SP**], where the analogous distances are 253.1 pm). As may be seen, the annelated [benzo] groups have relatively little effect on the degree of coplanarity of the chelate node. On the other hand, the size of the “chelate cell” in these macrocyclic compounds is large enough that, in principle, it could contain not only Be(II) but also ions of other metals, including M(II) ions of 3*d*-elements. For the non-bond angles (NNN) in the N3 grouping, within the framework of each of the calculation methods used by us, are equal to 60° in the case of [Be**SP**] and [Be**TBSP**] and are different in [Be**MBSP**] and [Be**DBSP**] (although this difference is very insignificant and, as a rule, does not exceed 0.5°). A similar situation also takes place for three six-membered metal chelates and five-membered non-chelate rings containing one N atom and four C atoms; in [Be**SP**] and [Be**TBSP**], all three cycles of both the first and second dimensions are the same, in [Be**MBSP**] and [Be**DBSP**], two are the same, while the third one differs from the other two in the sets of the bond angles and their sum. However, the degree of non-coplanarity of any of these rings is much less than that for chelate nodes since the sums of bond angles (BAS) in each of these structural fragments (BAS^6^ and BAS^5^, respectively) are very close to the values of 720.0° and 540.0°, corresponding to a flat hexagon and flat pentagon, respectively (the deviation of these sums from the indicated values is lesser than 2.5°). Taking into account all of the above, it can be argued that the molecular structures of each of the Be(II) complexes under study, calculated by both DFT methods used by us, reveal a very significant similarity to each other, both qualitatively and quantitatively.

The values of the electrical dipole moments for the [Be**SP**], [Be**MBSP**], [Be**DBSP**], and [Be**TBSP**] compounds calculated by both methods are presented in Table 2.

According to Table 2, the dipole moments differ markedly from zero, and this fact is associated with the absence of a center of symmetry in each of these complexes. The numerical values of this parameter calculated by these two DFT methods, as well as the parameters of molecular structures, do not generally differ too much from each other.

Key data of NBO analysis, and, namely, the values of effective charges on the central beryllium atom and the nitrogen atoms for the (666)macrotricyclic compounds under examination obtained by each of the DFT versions indicated above are presented in Table 3. Complete NBO analysis data for all these complexes are presented in Appendix A. In Table 3, the effective charges on beryllium and nitrogen atoms differ markedly from the close to integer values that should be characteristic of compounds with purely ionic chemical bonds. This fact, in our opinion, indicates a significant delocalization of the electron density in the coordination center of the complexes under study. The nitrogen atoms coordinating the Be atom have a significantly greater negative charge than the N1, N4, and N6 atoms in the meso-position. This charge distribution pattern is quite predictable since the difference in the electronegativity of the beryllium atom and its coordinating nitrogen atoms is significantly greater (1.34 and 3.04 on the Pauling scale) than the difference between the meso-nitrogen atoms and the associated carbon atoms (2.55 and 3.04).

At the same time, the charges on the key atoms indicated in Table 3 in the series [Be**SP**], [Be**MBSP**], [Be**DBSP**], and [Be**TBSP**] change insignificantly, but the dynamics of their changes for different atoms are different. Be is within the DFT B3PW91/TZVP and DFT M062X/def2TZVP methods, and there is a monotonic increase in the positive charge, while this dynamics for the nitrogen atoms has a more complex character and depends on the calculation method (Table 3).

Since Be(II) has a 1*s*^2^ electronic configuration, it is quite obvious that the ground state of this ion, as well as of the [Be**SP**], [Be**MBSP**], [Be**DBSP**], and [Be**TBSP**] complexes, which include it, must be a spin singlet (*M_S_*= 1), and the nearest excited state has an *M_S_* value that different from the spin multiplicity of the ground state (spin triplet) and should have much more energy. This prediction is supported by the <S**2> (operator of the square of the intrinsic angular momentum of the total spin of the system) values equal to 0.0000, which corresponds to the absence of unpaired electrons in each of these complexes and, consequently, the spin paired configuration 1*s*^2^ (Table 3). This conclusion is also in full agreement with the results of the NBO analysis within the framework of both calculation methods used by us, according to which the triplet state lies above the ground singlet state by more than 100 kJ/mol (see Appendix A). This circumstance argues in favor of the possibility of using calculations with a single-reference wave function. To check the wave functions of the ground and excited states for stability within the framework of both approximations of the DFT method, we used B3PW91/TZVP and M06/def2TZVP and used the standard procedure STABLE = OPT, which showed that the wave function for these states is stable with respect to the considered perturbations for the complexes studied in this work.

The images of the highest occupied and lowest vacant molecular orbitals (HOMO and LUMO, respectively) of the complexes under consideration, obtained using the DFT B3PW91/TZVP and DFT M06/def2TZVP methods, are shown in Figure 2 and Appendix A, respectively. As can be seen, the HOMO and LUMO shapes calculated by these variants of the DFT method are quite similar to each other, while their energies differ quite significantly from each other. At the same time, which is typical, during the transition from [Be**SP**] to [Be**TBSP**], i.e., with an increase in the number of annelated [benzo] groups, in general, there is an increase in the energies of both HOMO and LUMO. It is obvious that this is the result of some specific interactions between atoms, but the question of their nature is still open.

The standard thermodynamic parameters of formation (Δ*_f_H*^0^, *S*^0^, and Δ*_f_G*^0^) for the beryllium complexes under examination, obtained by the DFT B3PW91/TZVP method, are shown in Table 4. Of the two DFT approximations used in this work to calculate the thermodynamic characteristics Δ*_f_H*^0^, *S*^0^, and Δ*_f_G*^0^, we only used DFT/B3PW91/TZVP, despite M062X being a global hybrid functional with 54% HF exchange and being one of the best in performance within functional 06 for thermochemistry, kinetics, and non-covalent main group element interactions. However, this function should be used with caution when calculating the thermodynamic characteristics of organometallic compounds [26].

As can be noted, the value of both Δ*_f_H*^0^ and Δ*_f_G*^0^ is positive for each of these coordination compounds. This means that none of them can be obtained from the simple elements C, N, H, and Be under conditions of thermodynamic equilibrium. However, quantum chemical calculations carried out by the two DFT methods mentioned above show that each of the four macrocyclic beryllium compounds considered in this work is capable of existing as an individual chemical compound in the gas phase. At the same time, which is remarkable, in the series [Be**SP**]–[Be**MBSP**]–[Be**DBSP**]–[Be**TBSP**], the Δ*_f_H*^0^ and Δ*_f_G*^0^ values increase, so it can be assumed that with an increase in the number of annelated benzo groups, the resistance to decomposition into simpler components will slightly decrease.

## 3. Method

The DFT/B3PW91 method [27,28,29] combined with the standard extended split valence basis set TZVP was successfully used in [30,31,32]. According to [27,28,29], the calculations at this theoretical level make it possible, as a rule, to obtain values of geometric structural parameters that are close to experimental values, as well as thermodynamic characteristics that are acceptable in accuracy compared to other variants of the DFT method. To confirm the stability of the resulting solutions, we also used another relatively new version of the DFT method, M062X/def2TZVP, described in [33,34], which is most adequate for describing the parameters of the molecular and electronic structures of various *s-*, *p-,* and *d*-element compounds. This version of the DFT method, in principle, should provide more accurate data on the parameters of molecular and electronic structures than DFT B3PW91/TZVP; however, it is much more time consuming compared to it, and that is why we used it to compare the above data. Such a comparison was made by us in a recent article [32] using the example of Fe(VII) and Mn(VII) complexes with a 12-atomic (NNNN) tetradentate ligand and a nitride anion. The calculations were carried out using the Gaussian09 program package [35]. The correspondence of the found stationary points to the minimum on potential energy surfaces (PESs) was checked by calculating the vibrational frequencies. The optimized structures corresponding to the lowest total energy were selected for further consideration. NBO analysis was carried out using the methodology [36]. The standard thermodynamic parameters of formation Δ_*f*_*H*^0^, *S*^0^, and Δ_*f*_*G*^0^) for the [Be**SP**], [Be**MBSP**], [Be**DBSP**], and [Be**TBSP**] complexes under study were calculated according to the methodology described in [37].

## 4. Conclusions

As can be seen from the above, the data obtained by us using two different DFT methods with B3PW91/TZVP and M062X/def2TZVP theory levels quite reliably confirm the fundamental possibility of the existence of four new macrocyclic beryllium(II) compounds with doubly deprotonated forms of subporphyrazine ((H_2_**SP**)), mono[benzo] subporphyrazine (H_2_**MBSP**), di[benzo]subporphyrazine ((H_2_**DBSP**)), and tri[benzo]subporphyrazine (subphthalocyanine) (H_2_**MBSP**) with a ratio of Be(II) ion/ligand = 1:1. According to the results of these calculations, trigonal–pyramidal coordination of donor nitrogen atoms with respect to the Be(II) ion occurs in each of these complexes, with a very significant (almost 30°) deviation of the BeN_3_ chelate node from coplanarity; however, the six-membered metal chelate rings and the adjacent five-membered non-chelate rings are essentially coplanar. Characteristically, the presence of annelated [benzo] groups has practically no effect on the parameters of the molecular structure of the coordination center of the complexes. At the same time, what is important is that the molecular and electronic structures of the frame of the [Be**SP**], [Be**MBSP**], [Be**DBSP**], and [Be**TBSP**] obtained using the DFT B3PW91/TZVP and DFT M062X/def2TZVP methods practically coincide with each other, both qualitatively and quantitatively.

To date, there is no information in the literature about the synthesis of these complexes, and the point now is to obtain these unknown compounds in a real chemical experiment. In such a case, the prediction of the possibility of the existence of these coordination compounds and the modeling of their molecular structures using modern quantum chemical calculations (and, in particular, DFT methods of various levels) is a very useful tool in solving problems associated with such a synthesis. In this regard, it should be noted in conclusion that, according to the results of our calculations, the size of the chelate cell (cavity) of each of these ligands (in the representation that it has a spherical shape, its diameter is about 150 pm) is quite sufficient or slightly better and could also accommodate ions of some other *p*- and *d*-elements.

## Data Availability

No unpublished data were created or analyzed in this article.

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
