# Peer review of "First Examples of s-Metal Complexes with Subporphyrazine and Its Phenylene-Annulated Derivatives: DFT Calculations"

_ijms, 2024, doi:10.3390/ijms25136897_

Round 1

Reviewer 1 Report

Comments and Suggestions for Authors

My only advice to the authors is to also try a range-separated hybrid like wB97XD or CAM-B3LYP. The functionals they are using have a tendency to overly delocalize electrons, and may give spuriously large charge transfer in some cases. I would consider this an minor, optional, but important revision.

Comments on the Quality of English Language

It's fine. It's not a riveting read but it's well-written enough.

Author Response

Dear Reviewer,

We are deeply grateful to you for your work on our manuscript.

Please find below the responses to your comments.

Comments and Suggestions for Authors
Comment 1. My only advice to the authors is to also try a range-separated hybrid like wB97XD or CAM-B3LYP. The functionals they are using have a tendency to overly delocalize electrons, and may give spuriously large
charge transfer in some cases. I would consider this an minor, optional, but important revision.
Response. At one time, we tried to carry out calculations of 3d-metal complexes using methods that take into account dispersion interactions, the same wB97XD, as well as in the EmpiricalDispersion=GD3 variant. However, we are very cautious about the results of such calculations, since according to our data and the data of other authors (see, for example, in DOI 10.1134/S1070328421030064), in some cases this can lead to a distortion of the chelate center and an excessive decrease in bond lengths. But nevertheless, we always try to check the calculation taking into account dispersion interactions.

Comments on the Quality of English Language
It's fine. It's not a riveting read but it's well-written enough.

On behalf of authors,
Yours sincerely,
Prof. Dr. Georgiy V. Girichev

Reviewer 2 Report

Comments and Suggestions for Authors

The article is part of a wider series of investigations into molecular structures based on the porphyrazine molecule and its derivatives in the complexation of some metals. The studies presented here are purely theoretical and focus on determining the possibility of forming complex structures with metals that have a larger radius than the internal ring of the ligand.

Some observations and questions for the authors:
- What are the selection criteria for the DFT functionals used?
- Apart from the theoretical purpose of studying these structures, is there a practical one?
- The structure of the porphyrazine derivative, subporphyrazine, being non-planar and pyramidal-trigonal at the bonding atoms, reduces the importance of comparing the size of the metal radius relative to that of the internal ring of the ligand. This structure allows for the complexation of metals with a larger radius than a planar structure would permit.
- On page 2, before and after Scheme 1, "its" is repeated.
- On page 3, line 100, words written in Cyrillic letters, which are not in English, are used.
- For ease of comparison and tracking changes in structural and reactivity properties, I recommend using a notation for the atoms in structure b from Figure 1, [BeMBSP], similar to the others.
- "Calculated by DFT" in Table 1 can be omitted.

Comments on the Quality of English Language

- I recommend revising and improving the English language used.

Author Response

Authors’ Responses to Reviewer 2’ Report

Dear Reviewer,

We are deeply grateful to you for your work on our manuscript.

Please find below the responses to your comments.

All changes to the text made in accordance with the comments are highlighted in blue. 

Comments and Suggestions for Authors
The article is part of a wider series of investigations into molecular structures based on the porphyrazine
molecule and its derivatives in the complexation of some metals. The studies presented here are purely
theoretical and focus on determining the possibility of forming complex structures with metals that have a
larger radius than the internal ring of the ligand.

Some observations and questions for the authors:
Comment 1. What are the selection criteria for the DFT functionals used?
Response. The choice of the B3PW91 method was based on the data of the article DOI 10.1126/science.aah5975 and our own long-term experience in calculating 3d-metal complexes; the peculiarity of this method is that the
calculation accuracy is systematically improved as the basis set is expanded. The choice of the second method M062X is based on the fact that this method is specially developed for better calculation of 3dmetal complexes, and we already have quite a lot of our own positive experience in using this functionality.
Comment 2.  Apart from the theoretical purpose of studying these structures, is there a practical one?
Response. In fact, this work is aimed at establishing the fundamental possibility of the existence of metal complexes with subporphyrazine and its [benzo]annelated derivatives, which have not yet been detected in experiments. The synthesis of such compounds seems advisable at least because its implementation would serve as a certain contribution to the coordination chemistry of beryllium, which has not yet been fully studied. It is difficult at the moment to assess the possibilities of using such compounds, if they are synthesized, in any specific areas of anthropogenic activity.
Comment 3.  The structure of the porphyrazine derivative, subporphyrazine, being non-planar and pyramidal-trigonal
at the bonding atoms, reduces the importance of comparing the size of the metal radius relative to that of
the internal ring of the ligand. This structure allows for the complexation of metals with a larger radius than a planar structure would permit.
Response. We basically agree with this opinion of our esteemed Reviewer 2, but we must repeat once again that
despite the possibility of the existence of the chemical metal complexes considered in our article, which we
have shown, they have not yet been obtained experimentally. On the other hand, as can be seen from the data presented in Table 1, the lengths of the Be–N bonds, as well as the degree of deviation of the BeN3
metal chelate unit from coplanarity in the series [BeSP] – [BeMBSP] – [BeDBSP] – [BeTBSP] practically do
not change, which confirms the opinion expressed by our esteemed Reviewer. Since the above-mentioned
parameters of the structure of the complexes considered by us are among the key ones, their presence in Table 1 seems absolutely necessary to us.
Comment 4.  On page 2, before and after Scheme 1, "its" is repeated.
Response. We agree with this comment of our esteemed Reviewer, and we have made the corresponding correction in the revised text of the manuscript.
Comment 5.   On page 3, line 100, words written in Cyrillic letters, which are not in English, are used.
Response. We agree with this comment of our esteemed Reviewer 2, too, and we have made the corresponding
correction in the revised text of the manuscript.
Comment 6.   For ease of comparison and tracking changes in structural and reactivity properties, I recommend using a notation for the atoms in structure b from Figure 1, [BeMBSP], similar to the others.
Response. Frankly speaking, the meaning of this remark by our esteemed reviewer remained unclear to us, since the
numbering of the atoms in the chelate unit of BeN3 of the [BeMBSP] complex is the same as in the [BeSP],
[BeDBSP] and [BeTBSP] complexes, and if we rotate its structure, shown in Fig. 1b, by 120° counterclockwise, then the visual arrangement of the nitrogen atoms N2, N3 and N5 will be the same as in the other complexes.
Comment 7.  "Calculated by DFT" in Table 1 can be omitted.
Response. Also, We agree with this comment of our esteemed Reviewer 2, and we have made the corresponding correction in the revised text of the manuscript.

Comments on the Quality of English Language
Comment 8.  I recommend revising and improving the English language used.
Response. Unfortunately, our esteemed Reviewer 2 did not provide at least one or two examples of incorrect English words and/or phrases that he found in the text of the article; nevertheless, we made an attempt at the
revision he indicated.

On behalf of authors,
Yours sincerely,
Georgiy Girichev
